# A drop in Sahara dust fluxes records the northern limits of the African Humid Period

Daniel Palchan [1,2] & Adi Torfstein [1,2]

Northern and eastern Africa were exposed to significantly wetter conditions relative to present during the early Holocene period known as the African Humid Period (AHP), although the latitudinal extent of the northward expansion of the tropical rain belt remains poorly constrained. New records of $^{230}Th_{xs}$-normalized accumulation rates in marine sediment cores from the Red Sea and Gulf of Aden are combined with existing records of western Africa dust and terrestrial records across the Sahara Desert, revealing that fluxes of dust transported east from the Sahara decreased by at least 50% during the AHP, due to the development of wetter conditions as far north as ~22°N. These results provide the first quantitative record of sediment and dust accumulation rates in the Red Sea and the Gulf of Aden over the past 20 kyrs and challenge the paradigm of vast vegetative cover across the north and northeastern Sahara Desert during the AHP.

[1] The Fredy & Nadine Herrmann Institute of Earth Sciences, The Hebrew University of Jerusalem, The Edmond J. Safra Campus, Givat Ram, Jerusalem 91904, Israel. [2] Interuniversity Institute for Marine Sciences, Eilat 88103, Israel. Correspondence and requests for materials should be addressed to D.P. (email: Daniel.palchan@mail.huji.ac.il)

The primary source of dust to Earth's atmosphere is the Sahara Desert in northern Africa with an estimated annual flux of up to 1600–1700 Tg[1]. Atmospheric dust is strongly linked with global climate and environmental conditions through various mechanisms, including the scattering and absorption of solar and terrestrial radiation, modification of cloud properties, and fertilization of oligotrophic oceans with a subsequent enhancement of marine photosynthesis, which in turn modulates marine carbon uptake and atmospheric $CO_2$ concentrations[2,3]. It is well established that during the early-mid Holocene, wetter conditions than today prevailed in northern Africa because of the intensification and northward expansion of the African summer monsoon rains[4,5] triggered by both external changes in insolation flux and internal feedbacks from albedo variations over land[6–9]. Yet, the extent of the latitudinal expansion of the African summer monsoon precipitation is still debated[10]. Some studies support more-conservative estimations[11–13], whereas others suggest a green Sahara with vast vegetation cover, large waterbodies, rivers, and tropical rainfall that persisted in the currently hyperarid region[14–18]. Indeed, pollen records suggest that the Sahara Desert (north of 25°N) accommodated some tropical plants, but only along perennial rivers and waterbodies[13,19]. Accordingly, these do not necessarily reflect regional climate conditions but rather, the migration of some plants through designated waterways. Recent compilations of Holocene radiocarbon ages from terrestrial sites in the Sahara Desert, including lacustrine and palustrine environments[5,8,9] along with vegetation reconstructions[11], suggest these records are insufficiently resolved or discontinuous due to erosion and other hiatuses[13,20]. The influence of the intensification and migration of the monsoon rains was also recorded along the western margins of Africa in well dated, continuous marine sediment cores[7,21] as well as a lacustrine sediment core[20]. Multiple studies of riverine and windblown sediments from the marine environment at the Atlantic Ocean suggest that more-humid conditions prevailed through the Sahara and the Sahel between ~ 12 and 5 ka[4,7,20] with an abrupt termination and Saharan aridification at ~ 5.5 ka[4,8,20,22]. Dust emissions westward from the Sahara, co-varied with these environmental changes, where lower fluxes characterize the humid period[4,21,23,24]. Mediterranean sediment cores recorded the AHP environmental impact on dust as a shift in the kaolinite/chlorite clay proportions, which was interpreted to reflect extensive lake cover across northern Sahara[16]. Furthermore, a 3 Ma long record of eastern Mediterranean sediment (core 967) displays lower hematite concentrations in sapropel layers, including during the Holocene AHP interval. This is interpreted to reflect decreased dust flux owing to the greening of the Sahara[18]. Marine records from the Gulf of Aden[25] and from the western Arabian Sea[26] present an overall similar shift to wetter and dryer conditions at the early and mid Holocene, respectively. The latter observations, pertaining to conditions in eastern Africa are corroborated by a record of Nile Delta sediments, which reflects the climate patterns in the eastern Africa headwaters of the Nile River[27]. The northeastern flanks of Africa remain poorly studied in the context of the environmental impact of the AHP apart of some controversial evidence for carbonate deposition in terrestrial sites[28], which will be discussed below. It is worth mentioning, however, the recent study of speleothems from northeastern Africa[29], where episodes of growth occurred during previous interglacials but not during the Holocene. These carbonate deposits are further characterized by distinctly light oxygen isotopes, which were interpreted to reflect a northward expansion of summer African monsoon rains that brought Atlantic Ocean moisture to the Desert[29] rather than a proximal source in the Mediterranean Sea.

Here, we focus on the northeastern part of Africa and present new records of $^{230}Th_{xs}$-normalized dust accumulation rates (see

Methods) over the last 20 kyrs between the Gulf of Aden, central Red Sea, and northern Red Sea (Fig. 1; cores KL15; KL11; and KL23, respectively; for exact locations see Table 1). We combine the new dust accumulation rates with additional dust and hydrology records from sites in the Red Sea, the Atlantic Ocean and the Sahara Desert, to elucidate the latitudinal extent of the AHP impact in northern Africa and the eastern dust emissions during the last deglacial and Holocene. The results indicate a relatively limited northern extent of the AHP in eastern Africa (~ 22°N), and hence, challenge the paradigm of a vast vegetative cover across the north and northeastern Sahara Desert during the AHP.

## Results

**Dust sources and eastward transport.** Considering that the drainage basins that surround the Red Sea are relatively small, and given the regional hyperarid conditions that limit direct fluvial contributions to be negligible relative to the desert dust plumes[30], the terrigenous fraction in the Red Sea bottom sediments is considered to be overwhelmingly of eolian origin. The sources of the dust were identified using backward trajectory analyses (Supplementary Note 1): dust reaching the northern Red Sea typically originates from northern Libya and Egypt, dust reaching the central Red Sea originates from both Sudan and the Afar region, and dust reaching the Gulf of Aden is delivered from the Horn of Africa. Only a small fraction of the air parcels that are transported toward the Red Sea and the Gulf of Aden originate from the Arabian Peninsula (Supplementary Fig. 1). The provenance of the terrigenous fraction of downcore records in the Red Sea, based on their radiogenic isotope composition, confirms that the latter dust sources were similarly active during the late Quaternary and the Holocene[30].

Through the last deglaciation, dust accumulation rates in the Gulf of Aden and the central Red Sea dropped from maximum values of $0.72 \, g \, cm^{-2} \, ka^{-1}$ and $0.89 \, g \, cm^{-2} \, ka^{-1}$ during the deglacial to minimum values of $0.40 \, g \, cm^{-2} \, ka^{-1}$ and $0.35 \, g \, cm^{-2} \, ka^{-1}$ at ca. 7 ka, respectively (Figs. 1, 2a & Table 2). This ~twofold drop in dust accumulation rates is too big to be attributed to possible effects of grain size sorting on $^{230}Th_{xs}$-normalized fluxes[31], and is not the result of dilution of the dust component in the core sediments (i.e., by enhanced deposition of marine microfossils) as the $^{230}Th_{xs}$-normalized dust fluxes are relatively insensitive to this[32]. It is much more reasonable to attribute this change, which correlates with the monsoon index that is a measure of the intensity of the African Monsoons[33] (Fig. 2b), to changes in environmental conditions in the region that suppressed dust uptake from the source regions (Supplementary Note 1). In addition, all three cores display stable and close to 1 focusing factors (Supplementary Fig. 2), suggesting the calculated dust fluxes reflect vertical fluxes with minor effects of sediment focusing or winnowing (see Methods).

**Additional environmental proxies.** Records from the Nile Delta and from the Gulf of Aden reflect wetter conditions at eastern Africa during the AHP. Enhanced Nile River outflow is evident from the increased presence of basaltic detritus and depleted Mediterranean foraminifera $\delta^{18}O$ values (Fig. 2c) in Nile Delta core MS27[27] (Fig. 3), and a leaf wax $\delta D$ record from core RC09 from the Gulf of Aden[25] (Fig. 2i). These records are well correlated and point to a coeval precipitation increase in the Horn of Africa and its vicinity. The increased precipitation resulted in the enhanced growth of the vegetation cover[34] that inhibited dust uptake, a connection that is well expressed by the positive correlations between leaf wax $\delta D$ and dust accumulation rates in the Gulf of Aden ($R^2 = 0.3$) and the Central Red

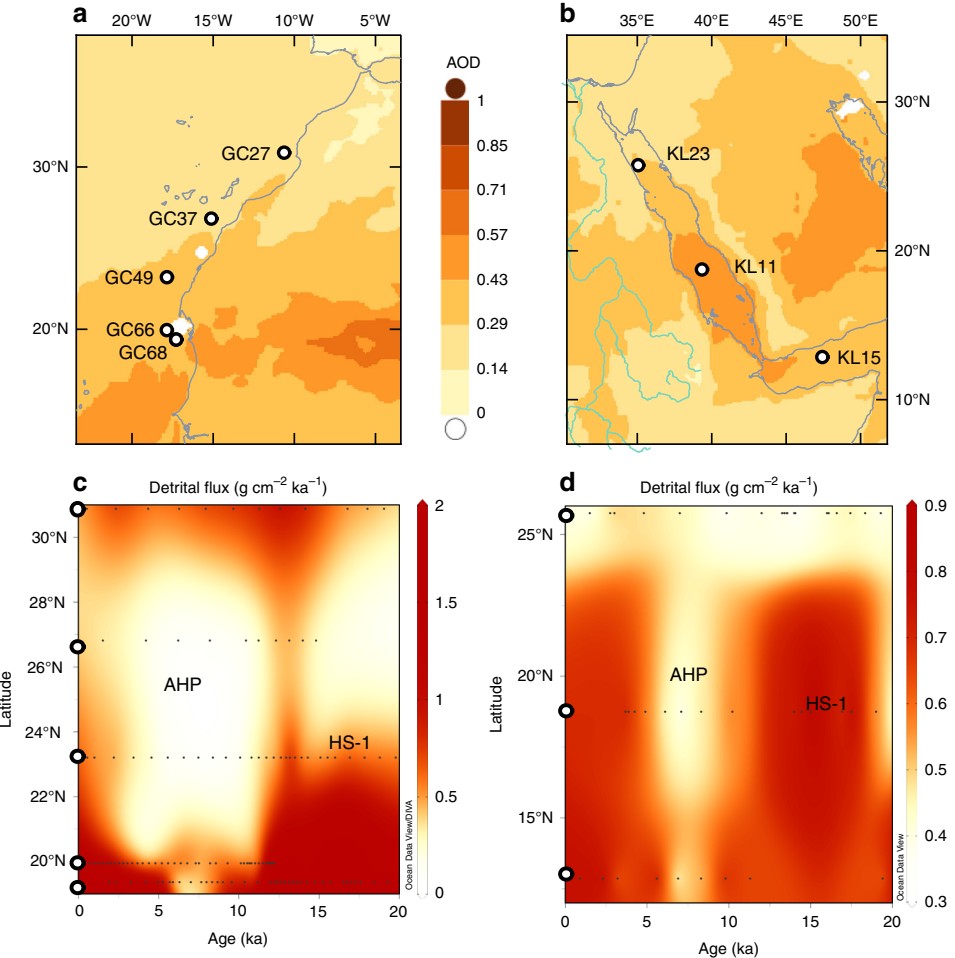

**Fig. 1** Spatiotemporal distribution of dust accumulation rates along the eastern and western margins of the Sahara Desert. **a** Location map for the Atlantic/Western Africa core sites with the aerosol optical depth (AOD; wavelength of 555 nm) averaged between 2000 and 2017, daily resolution on a 0.5° (created using Giovanni online data), and **c** corresponding dust accumulation rates through the last deglaciation[21] (white circles mark the latitudes of the cores). The impact of the African Humid Period (AHP) on dust accumulation in the Atlantic Ocean is evident mainly from the accumulation rates of the eolian fraction between latitudes 19°N to 28°N. North of 28°N, the data do not support lower dust fluxes during the AHP. **b** Red Sea-Gulf of Aden core sites and **d** corresponding dust accumulation rates. In the Red Sea and Gulf of Aden, higher and lower rates are observed during HS-1 and the AHP, respectively. Heinrich Stadial 1 (HS-1) is characterized by higher dust accumulation rates interpreted to reflect stronger wind intensities along with increased aridity in the source regions. The effect of HS-1 in both eastern and western Sahara extends up to ~ 24°N

**Table 1 Site locations**

| Region | Site | Latitude | Longitude | Reference |
|---|---|---|---|---|
| Red Sea and the Gulf of Aden | KL15 | 12° 51′30″N | 47° 25′54″E | This study |
| | KL11 | 18° 44′30″N | 39° 20′36″E | This study |
| | KL23 | 25° 44′54″N | 35° 3′18″E | This study |
| | KL09 | 19° 57′36″N | 38° 8′18″E | Roberts et al.[36] |
| | RC09 | 12° 2′ 30″ N | 44° 0′ 40″ E | Tierney et al.[7,25] |
| | Wadi Sannur | 31° 17′N | 28° 37′E | El-Shenawy et al.[29] |
| Arabian Sea | 93KL | 23° 35′ N | 64° 13′ E | Pourmand et al.[37] |
| Atlantic sector | GC68 | 19° 21′47″ N | 17° 16′55″W | McGee et al.[21] |
| | GC66 | 19° 26′38′ N | 17° 51′36″ W | McGee et al.[21] |
| | GC49 | 23° 12′ 22″ N | 17° 51′14″ W | McGee et al.[21] |
| | GC37 | 26° 48′58″ N | 15° 7′4.8″ W | McGee et al.[21] |
| | GC27 | 30° 52′48″ N | 10° 37′48″ W | McGee et al.[21] |
| | Grote de Piste | 33° 50′ 24″ N | 4° 5′ 24″ W | Wassenburg et al.[39] |
| Mediterranean | 293G | 36° 10.414′N | 2° 45.280′W | Rodrigo-Gámiz et al.[17] |
| | SL71 | 34° 48.67′N | 23° 11.65′E | Ehrmann et al.[16] |
| | MS27 | 31° 47.90′N | 29° 27.70′E | Revel et al.[27] |
| | 967 | 34° 04′N | 32° 43′E | Larrasoana et al. 2003 |

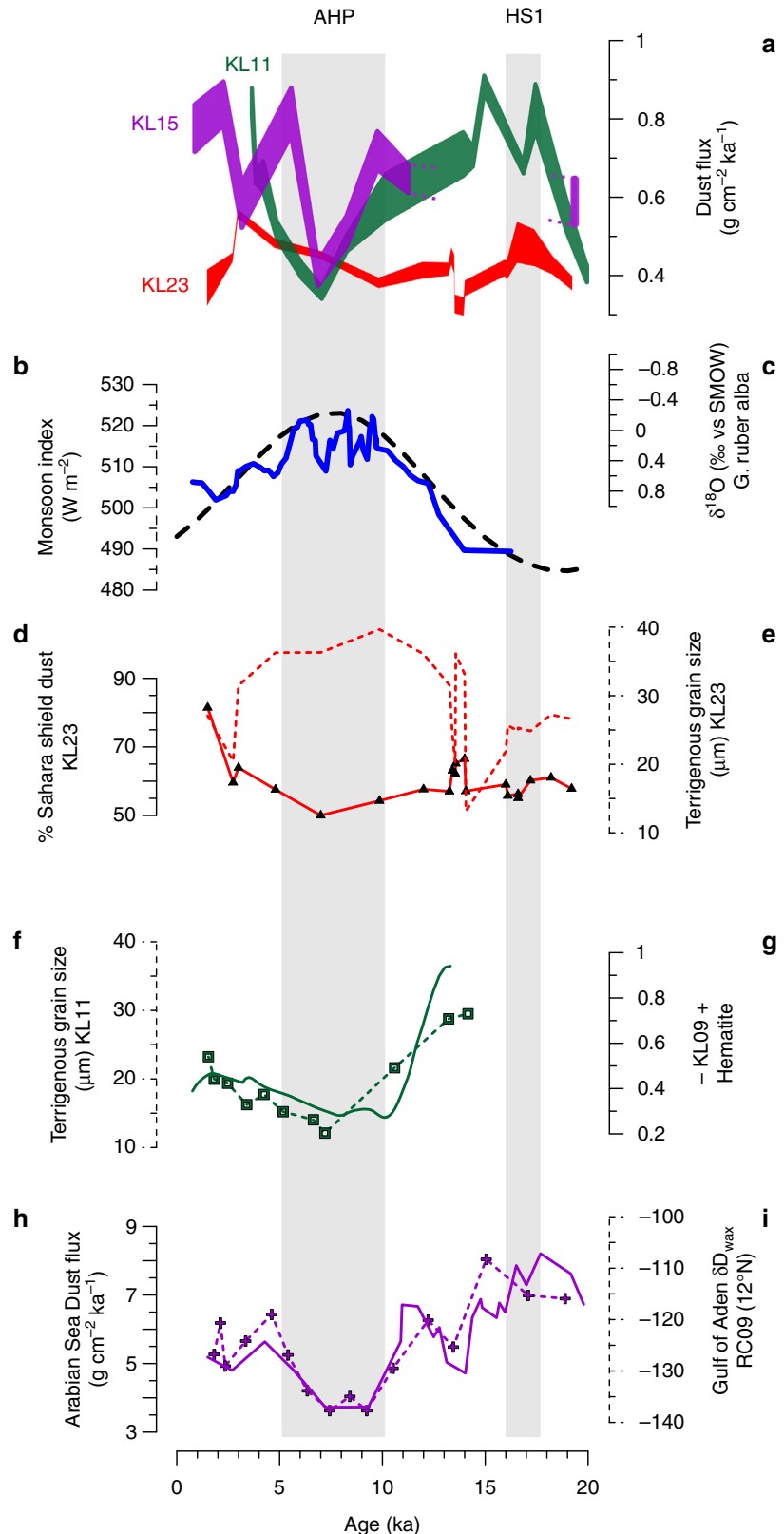

Sea ($R^2 = 0.6$) (Supplementary Note 1). This relationship suggests that over millennial time scales, increased precipitation (characterized by lower leaf wax δD[25]) is associated with lower dust fluxes, a correlation that is also established on a decadal timescale for western Africa[35]. In addition, hematite concentrations in core KL09 in the central Red Sea, proximal to

core KL11 (Fig. 3), were interpreted to serve as a proxy for eolian dust, hence indicating a drop in dust fluxes during the early Holocene[36] (Fig. 2g). A drop in dust accumulation rates during the AHP is also recorded further to the east, in the Arabian Sea (~ 22°N), where dust accumulation rates dropped by ~ 50% relative to the last glacial[37] (Fig. 2h). Within the Red

**Fig. 2** East Africa environmental proxies. **a** $^{230}Th_{xs}$ -normalized dust mass accumulation rates; envelopes represent a 2σ uncertainty. **b** Monsoon index, calculated from the summer (August) insolation as Monsoon index = Insol.(C) + (Insol. (C)−Insol. (E)), where (C) represents the tropic of Cancer (~ 23°N) and (E) represents the equator[59]. **c** Planktonic foraminifera $\delta^{18}O$ (five point running average) in core MS27 (Nile Delta); lighter $\delta^{18}O$ values reflect enhanced flow of Nile River freshwater into Mediterranean waters[27], **d** Saharan dust fraction (%) comprising the eolian sediments in core KL23. **e** Grain size mode of the siliciclastic fraction in core KL23. **f** Grain size mode of the siliciclastic fraction in core KL11. **g** Hematite concentrations, reflecting dust contents, in core KL09[36]. **h** $^{230}Th_{xs}$-normalized dust mass accumulation rates at core 93KL[37] (Arabian Sea). **i** $\delta D_{wax}$ values from core RC09 in the Gulf of Aden, where low and high values reflect more and less rainfall, respectively, in the Horn of Africa and its vicinity, the Ethiopian Highlands[25]. The observations indicate wetter conditions and lower dust accumulation rates during the African Humid Period (AHP) persisted as far north as the central Red Sea (~ 22°N) whereas the northern Red Sea shows a negligible change during the AHP. Higher dust fluxes observed for Heinrich Stadial 1 (HS-1)

### Table 2 Sediment mass accumulation rates in the Red Sea and Gulf of Aden

| Core | Depth (cm) | Age (ka) | Water depth (m) | $^{232}Th$ (dpm g$^{-1}$) | 1σ | $^{230}Th$ (dpm g$^{-1}$) | 1σ | $^{238}U$ (dpm g$^{-1}$) | 1σ | Bulk flux (g cm$^{-2}$ ka$^{-1}$) | 1σ | Terrigenous flux (g cm$^{-2}$ ka$^{-1}$) | 1σ |
|---|---|---|---|---|---|---|---|---|---|---|---|---|---|
| KL15 | 0 | 0.9 | 1628 | 0.947 | 0.002 | 2.243 | 0.011 | 2.422 | 0.004 | 2.60 | 0.08 | 0.77 | 0.03 |
| KL15 | 10 | 2.3 | 1628 | 0.981 | 0.001 | 2.179 | 0.009 | 1.553 | 0.003 | 2.71 | 0.07 | 0.84 | 0.03 |
| KL15 | 16 | 3.2 | 1624 | 0.802 | 0.002 | 2.413 | 0.013 | 3.570 | 0.005 | 2.29 | 0.09 | 0.58 | 0.03 |
| KL15 | 32 | 5.6 | 1632 | 0.960 | 0.002 | 2.239 | 0.009 | 3.183 | 0.004 | 2.70 | 0.08 | 0.82 | 0.03 |
| KL15 | 41 | 6.9 | 1625 | 0.545 | 0.001 | 2.445 | 0.010 | 6.413 | 0.003 | 2.34 | 0.07 | 0.40 | 0.02 |
| KL15 | 51 | 8.3 | 1615 | 0.699 | 0.001 | 2.458 | 0.011 | 4.712 | 0.003 | 2.32 | 0.07 | 0.51 | 0.02 |
| KL15 | 61 | 9.8 | 1604 | 0.644 | 0.002 | 1.734 | 0.011 | 3.188 | 0.005 | 3.54 | 0.09 | 0.72 | 0.02 |
| KL15 | 71 | 11.3 | 1604 | 0.685 | 0.001 | 2.072 | 0.008 | 4.149 | 0.002 | 2.99 | 0.06 | 0.64 | 0.02 |
| KL15 | 103 | 19.4 | 1544 | 1.001 | 0.001 | 2.804 | 0.010 | 2.727 | 0.003 | 1.86 | 0.07 | 0.59 | 0.03 |
| KL11 | 25 | 3.7 | 821 | 0.721 | 0.001 | 1.018 | 0.005 | 0.870 | 0.002 | 3.71 | 0.059 | 0.84 | 0.02 |
| KL11 | 27 | 3.9 | 814 | 0.450 | 0.000 | 0.748 | 0.008 | 0.814 | 0.001 | 4.57 | 0.038 | 0.65 | 0.01 |
| KL11 | 31 | 4.2 | 822 | 0.584 | 0.001 | 0.974 | 0.007 | 1.065 | 0.005 | 3.54 | 0.082 | 0.65 | 0.02 |
| KL11 | 38 | 4.9 | 815 | 0.431 | 0.001 | 0.859 | 0.005 | 1.398 | 0.003 | 3.78 | 0.063 | 0.51 | 0.01 |
| KL11 | 51 | 6.1 | 799 | 0.366 | 0.001 | 0.843 | 0.005 | 1.243 | 0.003 | 3.55 | 0.066 | 0.41 | 0.01 |
| KL11 | 60 | 7.1 | 771 | 0.334 | 0.001 | 0.855 | 0.004 | 1.433 | 0.002 | 3.34 | 0.056 | 0.35 | 0.01 |
| KL11 | 71 | 8.3 | 798 | 0.468 | 0.001 | 1.030 | 0.004 | 2.233 | 0.002 | 3.26 | 0.056 | 0.48 | 0.01 |
| KL11 | 80 | 10.2 | 748 | 0.875 | 0.001 | 1.659 | 0.008 | 3.849 | 0.004 | 2.17 | 0.082 | 0.60 | 0.03 |
| KL11 | 90 | 14.0 | 750 | 0.888 | 0.001 | 1.865 | 0.009 | 5.819 | 0.003 | 2.53 | 0.077 | 0.71 | 0.03 |
| KL11 | 98 | 14.5 | 749 | 0.391 | 0.001 | 0.809 | 0.005 | 2.376 | 0.003 | 5.69 | 0.077 | 0.70 | 0.01 |
| KL11 | 106.5 | 15.0 | 749 | 0.515 | 0.001 | 0.837 | 0.003 | 1.931 | 0.002 | 5.46 | 0.054 | 0.89 | 0.01 |
| KL11 | 117 | 16.9 | 748 | 0.307 | 0.001 | 0.645 | 0.002 | 1.677 | 0.002 | 6.95 | 0.055 | 0.67 | 0.01 |
| KL11 | 118.5 | 17.5 | 748 | 0.850 | 0.002 | 1.148 | 0.007 | 1.103 | 0.004 | 3.09 | 0.087 | 0.83 | 0.03 |
| KL11 | 131.5 | 19 | 747 | 0.709 | 0.001 | 1.211 | 0.005 | 1.100 | 0.003 | 2.47 | 0.072 | 0.55 | 0.02 |
| KL23 | 5 | 1.5 | 703 | 0.320 | 0.001 | 0.710 | 0.024 | 0.803 | 0.002 | 3.63 | 0.054 | 0.37 | 0.01 |
| KL23 | 10 | 2.7 | 702 | 0.410 | 0.002 | 0.790 | 0.005 | 0.703 | 0.004 | 3.43 | 0.080 | 0.44 | 0.01 |
| KL23 | 15 | 3.0 | 700 | 0.411 | 0.001 | 0.683 | 0.004 | 0.697 | 0.002 | 4.28 | 0.064 | 0.55 | 0.01 |
| KL23 | 20 | 4.8 | 695 | 0.475 | 0.001 | 0.861 | 0.005 | 0.796 | 0.002 | 3.20 | 0.062 | 0.48 | 0.01 |
| KL23 | 25 | 7.0 | 703 | 0.504 | 0.001 | 0.983 | 0.004 | 1.268 | 0.002 | 2.84 | 0.058 | 0.45 | 0.01 |
| KL23 | 30 | 9.9 | 675 | 0.382 | 0.002 | 0.858 | 0.006 | 1.434 | 0.003 | 3.14 | 0.080 | 0.38 | 0.01 |
| KL23 | 35 | 12.0 | 638 | 0.529 | 0.001 | 1.108 | 0.009 | 1.956 | 0.003 | 2.46 | 0.073 | 0.41 | 0.02 |
| KL23 | 40 | 13.3 | 629 | 0.787 | 0.001 | 1.606 | 0.007 | 2.572 | 0.002 | 1.67 | 0.063 | 0.41 | 0.02 |
| KL23 | 45 | 13.4 | 629 | 0.993 | 0.001 | 1.990 | 0.008 | 3.658 | 0.003 | 1.43 | 0.072 | 0.45 | 0.03 |
| KL23 | 50 | 13.5 | 629 | 1.137 | 0.001 | 2.377 | 0.009 | 4.528 | 0.003 | 1.19 | 0.072 | 0.43 | 0.03 |
| KL23 | 55 | 13.6 | 629 | 1.060 | 0.002 | 2.905 | 0.013 | 6.935 | 0.003 | 0.97 | 0.080 | 0.32 | 0.03 |
| KL23 | 60 | 14.0 | 626 | 1.062 | 0.002 | 2.806 | 0.013 | 5.768 | 0.004 | 0.95 | 0.082 | 0.32 | 0.04 |
| KL23 | 65 | 14.1 | 626 | 0.846 | 0.001 | 1.885 | 0.009 | 2.982 | 0.002 | 1.36 | 0.071 | 0.36 | 0.02 |
| KL23 | 70 | 16.0 | 626 | 1.034 | 0.001 | 2.039 | 0.008 | 2.722 | 0.002 | 1.28 | 0.072 | 0.42 | 0.03 |
| KL23 | 75 | 16.1 | 626 | 1.046 | 0.001 | 2.115 | 0.010 | 3.028 | 0.003 | 1.25 | 0.077 | 0.41 | 0.03 |
| KL23 | 80 | 16.6 | 626 | 1.098 | 0.004 | 1.992 | 0.018 | 2.798 | 0.004 | 1.39 | 0.105 | 0.48 | 0.05 |
| KL23 | 88 | 17.4 | 626 | 1.003 | 0.003 | 2.053 | 0.017 | 3.909 | 0.004 | 1.48 | 0.098 | 0.47 | 0.04 |
| KL23 | 94 | 18.3 | 615 | 0.541 | 0.002 | 1.263 | 0.009 | 2.774 | 0.003 | 2.48 | 0.081 | 0.42 | 0.02 |
| KL23 | 98 | 19.3 | 615 | 0.834 | 0.001 | 2.008 | 0.008 | 3.888 | 0.001 | 1.44 | 0.062 | 0.38 | 0.02 |

Sea, grain size fining of the terrigenous fraction has been shown to be associated with the occurrence of short fluvial episodes[30,38], even though these are currently extremely rare and quantitatively insignificant in this region. Accordingly, grain size fining during the AHP at KL11 (Fig. 2f; Supplementary Table 1) represents the triggering of local floods, probably originating from the Baraka Basin (Fig. 3) owing to increased precipitation south of 19°N. The partial accumulation of fluvial deposits at KL11 during the AHP, suggests that dust accumulation rates during this interval at the central Red Sea were even smaller than the a priori values reported above. Thus, the eastward dust fluxes dropped by > 50% during the AHP, perhaps even by 80% as reported for the westward Sahara dust plume[21] (Fig. 1a).

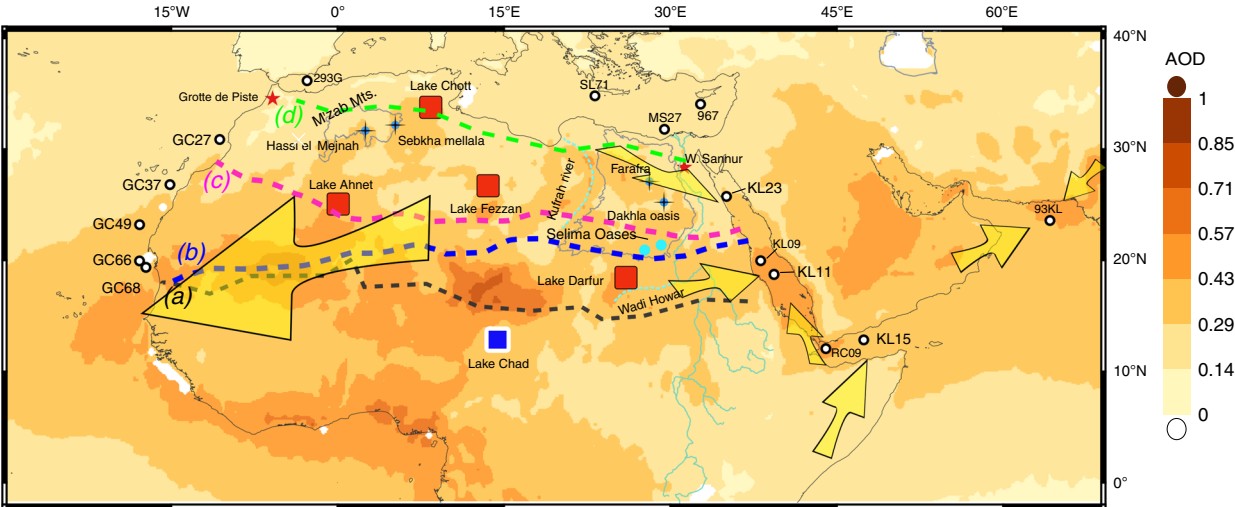

**Fig. 3** Climatic records of the African Humid Period (AHP). $^{230}Th_{xs}$-normalized dust accumulation rates in cores KL23, KL11, and KL15 from the Red Sea and Gulf of Aden. Dust accumulation rates and other properties from: the Atlantic Ocean GC27, GC37, GC49, GC66, and GC68 reported by McGee et al.[21]; Red Sea core KL09[36]; Arabian Sea RC09[25] and 93KL[37]; Mediterranean Sea 293G[17], SL71[16], MS27[27], and core 967[18]. Yellow arrows indicate the primary dust routes from land to the studied sites as computed from air mass back trajectories. Red stars locate the speleothem caves of Grotte de Piste[39], and Wadi Sannur[29], red squares indicate the approximate location of rejected mega lakes, and a blue square indicates the approximate location of mega lake Chad[10]. The bold dashed lines show various precipitation borders, note the letters on their left end: black **a** modern 50 mm/yr isohyat[60], blue **b** AHP modeled increase of precipitation by 50 mm/yr from PMIP3[10], purple **c** northern most reach of African monsoon precipitation as proposed here, and green **d** the approximate northern extent of evidence supporting wetter conditions during the AHP. The green line was drawn based on previous compilations from: lacustrine and palustrine deposits[8,15], vegetation based modeling[34], leaf wax data[7], mineralogical studies in the Mediterranean[16] and the suggested mega lakes[10]. Dust records from the northern Red Sea[30] indicate negligible environmental perturbations in this region during the AHP. Collectively, these observations indicate that monsoonal precipitation and the environmental change it drives (i.e., increased soil humidity, vegetation cover and dust retention) did not reach the latitudes from where dust originated to the northern Red Sea north of ~ 25°N. However, we draw the purple line even further to the south because the Farafra and Dakhla oases suggest that increased water availability at this time was driven by changes in groundwater discharge, rather than direct precipitation. These conclusions are further supported by pollen based reconstructions of soil humidity and precipitation[44]. The northwestern extent of the AHP impact area is reflected by dust accumulation rates in the Atlantic Ocean[21] where core GC27 (31°N) does not show a significant drop in dust accumulation rates

Surprisingly, dust accumulation rates in the northern Red Sea (KL23) display little change across the AHP (ca. $0.41 \pm 0.03$ g cm$^{-2}$ ka$^{-1}$; Fig. 2a). Other properties of the dust deposited in KL23 such as grain size and geochemical signature remain relatively constant as well (Fig. 2d, e; Supplementary Table 1). Assuming the siliciclastic dust in core KL23 is comprised of erosion products from both distal Saharan granitoids and proximal Arabian-Nubian shield granitoids, the εNd compositions measured along the core[30] allow to evaluate the relative fraction of distal contribution in the sediments (Supplementary Note 2). Accordingly, the fraction of the distal Saharan dust end member is < 5% smaller during the AHP relative to the late glacial period (Fig. 2e). Combined, these observations suggest that precipitation and dust uptake rates in its source regions of terrigenous material to the northern Red Sea remained relatively invariable across the AHP in the northern Sahara Desert (Fig. 2). Thus, the steady but low dust fluxes from the northern Sahara Desert constrain the northern limit of the environmental impact of the Holocene AHP in the Red Sea region to reach no further north than 24°N, in agreement with conservative PMIP3 climate model reconstructions[10] (Fig. 3).

## Discussion

To confine the northern limits of the monsoon expansion during the AHP we turn to examine terrestrial records across the Sahara Desert from west to east. At its northwestern edge, the speleothem cave record from Grotte de Piste (Fig. 3) indicates relatively stronger precipitation during the early Holocene[39]. However, model simulations suggest this is related to variations

in the North Atlantic sea level pressure rather than sub-equatorial insolation changes[39]. Nearby in the western Mediterranean, the εNd composition of terrigenous particles from sediment core 293 G do not display patterns corresponding with the AHP[17].

Sediment records from the west African sector of the Atlantic Ocean show a robust negative correlation between dust accumulation rates[21,24,32] and precipitation in western Africa[7]. This correlation, however, fades towards the north (core site GC27; Fig. 1), reflecting the fact that the δD$_{wax}$ record might be biased by the mountainous terrain of the Atlas Mts. where precipitation is tightly related to the North Atlantic during the early Holocene[39]. Accordingly, the northern extent of the monsoon expansion in western Africa during the AHP reached to ~ 28°N (Fig. 3). Further to the east, two sites—Hassi el Mejnah and Sebkha Mellala (Fig. 3)—display larger water volumes during the AHP[40,41]. These however, could be affected from a similar source of precipitation as the Grotte de Piste record, and probably also reflect a component of groundwater discharge (i.e., recharged at the M'zab mts. > 1000 m elevation) rather than direct precipitation[41]. Similarly, a recent compilation of records studied across the central Sahara Desert[20], suggests wetter conditions existed during the AHP, yet many of these archives record groundwater activity rather than direct precipitation[5]. Indeed, an investigation of the geomorphologic features of assumed paleo-mega lakes in the Sahara Desert (Lakes Chott, Fezzan, Ahmet, Darfur, and Chad; Fig. 3) had questioned the existence of four out of five mega lakes, and associated the related observations to a local groundwater regime that is disconnected from local precipitation[10]. Further to the east, two sites located west of the Nile Valley (i.e., Farfara and Dakhla oases; Fig.3) display a priori wetter

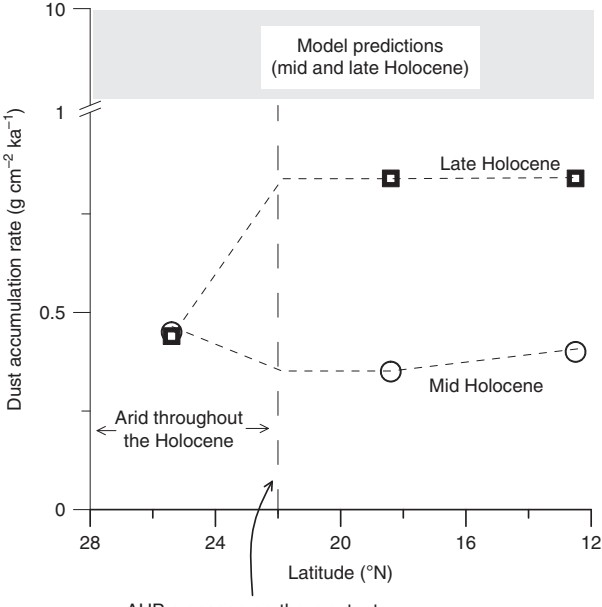

**Fig. 4** Dust accumulation rates in the eastern Sahara during the mid and late Holocene (6–7 and 2–4 ka, respectively) from models and $^{230}Th_{xs}$-normalized rates (this study). Model predictions[45,46] overestimate the observed values by an order of magnitude (note the break in the axis) and therefore do not capture the African Humid Period (AHP) shift in dust fluxes. Dust accumulation rates in the northern Red Sea along with environmental evidence suggest that the northeastern Sahara Desert was arid north of latitude ~22°N throughout the length of the Holocene

conditions during the AHP[20], yet they are supplied by groundwater whose recharge areas extend as far south as 20°N. Tuffas and travertines that were recently dated at these sites as well as elsewhere in the eastern Sahara Desert, indicate most of the sediments were deposited during glacial periods when the region was arid[42]. Accordingly, Abotalib et al.[42] postulates that these ages are associated with the time of groundwater discharge rather than the time of meteoric precipitation, which is supported by the large distance to the recharge areas of the aquifer in the south (100–500 km away) and the slow groundwater velocities (0.5–3.5 m/yr). Indeed, speleothems at Wadi Sannur (Fig. 3) that had grown during earlier interglacial peaks (e.g., MIS 9, 7, 5e) did not form during the Holocene, indicating that direct precipitation in this region during the AHP was very limited[29]. Therefore, the palustrine sediments in the northeastern Sahara do not necessarily reflect local precipitation but rather respond to hydrological changes related to distant monsoon activity in the south (Supplementary Note 3). However, high organic matter content in buried lake sediments along with abundant muds and indicative for freshwater diatom and pollen records from the Selima Oases (Fig. 3), point to increased precipitation during the AHP[43,44], suggesting that the limit of the AHP monsoon rains reached ~22°N at the northeastern Sahara Desert. This is in agreement with the new results of dust accumulation rates in the Red Sea and Gulf of Aden, which indicate that the northern extent of the AHP monsoon rains reached no further than 22°N. This suggests a minimal impact, if at all, of the monsoonal hydrological systems on eastward dust emissions out of northern Sahara. Hence, the northeastern extension of AHP humidity, which suppressed dust emissions by at least 50% between 10 and 5 ka, was constrained south of latitude ~22°N, in agreement with the reconstruction of steppe-desert ecotone[11]. Dust accumulation rates in the Atlantic Ocean, at the other side of the continent,

dropped by ~80% between 12 and 5 ka at 19°N[21], within the pathway of the modern western Saharan dust plume, compared to a ~25% drop at 27°N, north of the main dust plume (Fig. 1). These results reflect, in agreement with the established millennial inverse correlation between precipitation and dust fluxes (Supplementary Fig. 3), increased wetter conditions relative to present that existed in western Sahara up to ~28°N during the AHP[7,21] (Fig. 3 and see Supplementary Note 4 for a summary of the above mentioned observations).

Despite the similar temporal trends in westward and eastward Saharan dust emissions, these differ in both magnitude and latitudinal extent. Model evaluations of dust fluxes during the Holocene in eastern Africa are currently based on very few dispersed data points[45], where most are located in the Atlantic Ocean and few in the Arabian Sea[3,45,46]. Consequently, these models lack the sensitivity to capture eastward Saharan dust emissions. Recent compilations report an order of magnitude range in eastern African dust emissions during the Holocene, between 1 and 10 $g\,cm^{-2}\,ka^{-1}$ and do not account for the observed millennial-timescale variations in dust accumulation rates[45,46] (Fig. 4).

These latter values however are about an order of magnitude higher than the new results reported here in the Red Sea for the past 20 kyr (~0.4–0.9 $g\,cm^{-2}\,ka^{-1}$) and fail to capture significant differences between the mid Holocene and the late Holocene values, which reflect wetter conditions during the AHP (Fig. 4). The new findings can provide important constraints on the understanding of climate change patterns across northern Africa and should be used to tune climate models that account for the role of atmospheric dust in modulating global climate change.

## Methods

**Sediment cores**. The three cores studied here encompass the eastern borders of the Sahara and the Sahel (Fig. 1): KL23 in the northern Red Sea (25°N, water depth 702 m), KL11 in the central Red Sea (18°N, water depth 825 m) and KL15 in the Gulf of Aden (13°N, water depth 1631m). See Table 1 for exact locations. All cores were recovered during the METEOR cruise M 31/2 and 31/3 at 1995 and cruise M 5/2 at 1987[47].

The lithology of the cores comprises marine microfossils and fine detritus[47–49]. The sediments in the Red Sea cores show variations from bright yellow sediments that are rich with carbonates (up to 75%) and characterize interglacial intervals, with intercalating dark gray-colored horizons with very low carbonate content (<15%) characterizing glacial intervals. The fine detritus comprises rock dust primarily derived from the weathering of continental accumulation basins across the Sahara Desert and the Arabian Peninsula[30,36,49]. In the Red Sea, grain size distribution of the detrital fraction shows unimodal distribution during glacial intervals and an addition of a smaller grain size during interglacial intervals[30]. Detrital grain size finning in the Red Sea sediments was shown to characterize input of fluvial material, and hence, suggest wetter periods in the neighboring watersheds[30,38]. The sources of the dust were studied elsewhere[30] and are used here as well (see Supplementary Note 2).

The three studied cores have a detailed stable isotope record that was developed from the analysis of planktonic foraminifera (mainly G. ruber)[47]. A clear difference between the oxygen isotope values is seen between the open ocean environment (i.e., in KL15 the Gulf of Aden) and the enclosed basin conditions (i.e., KL11 and KL23 in the Red Sea). The δ18O composition ranges between −2 and 6 in the Red Sea and between −2 and 0 in the open ocean. The heavier values occur during glacial times and are caused by the increased salinities of the water column of the Red Sea owing to evaporation and the limited connection to the open ocean because of low sea level[47]. In fact, core KL23 shows heavier values during glacial intervals (up to 6‰), whereas the heaviest values observed in core KL11 are more moderate (up to 4‰). Accordingly, the δ18O stratigraphy of the Red Sea cores can be compared with the globally stacked δ18O curve, thereby providing additional chronological constraints[47,50]. High-resolution study of the δ18O curve of the Red Sea sediments allowed distinguishing acute climatic variations correlating these to the global climate assisting in understanding of the chronology of the Red Sea cores[51]. Recently, the record was further tuned to the high resolution and well dated (by U-Th) speleothem record of Soreq cave in Israel[51].

**Analytics**. Bulk sediment samples were digested using concentrated aqua regia followed by cycles of $HNO_3$-HF mixtures until full digestion was achieved. After drying, the samples were re-dissolved in 5 ml 4 N $HNO_3$. From these solutions, an

aliquot was extracted and diluted in preparation for major and trace element analysis using an Agilent 7500cx inductive coupled plasma mass spectrometer (ICP-MS) at the Hebrew University of Jerusalem. U and Th were purified from the residual solution through conventional column ion chromatography using AG1 X-8, 200–400 mesh anion resin. Prior to their analysis, the purified Thorium aliquots were doped with an appropriate amount of U to evaluate the mass bias during each measurement, which was corrected for using the exponential law. The samples were analyzed for their $^{230}Th$, $^{232}Th$, $^{235}U$, and $^{238}U$ content using a Neptune plus multi-collector inductive coupled plasma mass spectrometer (MC-ICP-MS) at the Institute of Earth Sciences, the Hebrew University of Jerusalem. Replicates of international standard IRMM-035 ($n = 75$) and a calibrated in-house Th standard ($n = 57$) were measured routinely to monitor the results. In addition, each batch of 10 samples included one full procedural blank and two international basalt standards; BCR-2 and BHVO-2. The basalt standards yielded $^{230}Th/^{232}Th$ activity ratios of: $0.880 \pm 0.005$ for BCR-2 ($n = 11$, $2\sigma$) and $1.090 \pm 0.008$ ($n = 6$, $2\sigma$) for BHVO-2. These values are in agreement with published values of $0.882 \pm 0.002$ for BCR-2 and $1.093 \pm 0.002$ for BHVO-2[52].

**$^{230}Th$-normalized accumulation rates and focusing factor**. $^{230}Th$ is produced by alpha decay of $^{234}U$ at a constant rate of 0.0267 dpm m$^{-3}$ yr$^{-1}$ [53]. Uranium is highly soluble in seawater with a relatively long residence time ($\sim 400$ kyr), resulting in its conservative behavior in the oceans. $^{230}Th$ concentrations, however, are four orders of magnitude lower than its parent U isotope, with a residence time ca. 20 yr owing to its efficient removal from the water column by adsorption onto settling particles (i.e., scavenging). Thus, virtually all the radiogenic Th produced in the marine environment is transported to the sea floors. The flux of $^{230}Th$ to the sea floor is modulated by the depth of the water column, and to a lesser extent water salinity. The constant production of $^{230}Th$ and its removal by settling particles results with a simple inverse relationship where higher fluxes result in lower $^{230}Th$ concentrations in the sediments and vice versa[53]:

$$PF_T = \frac{\beta \, x \, z}{^{230}Th_{xs}^0} \quad (1)$$

Where $PF_T$ is the total particle flux, $\beta$ is the production rate of $^{230}Th$, z the water depth and $^{230}Th_{xs}^0$ the activity in dpm g$^{-1}$ of $^{230}Th$ corrected for: (1) water depth (2) lithogenic input (3) authigenic input, and (4) the decay of $^{230}Th$. The Red Sea is a rather shallow basin relative to the typical deep water columns where this method has been previously applied[21,23,37,54], and hence, its water depth is sensitive to orbital-timescale sea level changes[51]. Indeed, water depths in the sites studied here varied by up to 15% during the past 20 ka. The lithogenic $^{230}Th$ was corrected for by assuming secular equilibrium between $^{238}U$ and $^{230}Th$, and a value of ($^{238}U/^{232}Th$) = 0.6 in the detrital fraction[55]. The fraction of authigenic U and Th in the samples was evaluated by subtracting the detrital $^{238}U$ from the total measured U, using the seawater ratio of $^{234}U/^{238}U = 1.148$ [56] in the seawater and correct for the decay of $^{234}U$ using $^{234}\lambda = 2.82 \times 10^{-6}$ yr$^{-1}$ [57]. Following these corrections, we calculated the initial excess $^{230}Th$ using $^{230}\lambda = 9.19 \times 10^{-6}$ yr$^{-1}$ [57]. The results are reported in Table 2, including the calculated $^{230}Th$-normalized mass accumulation rates of the bulk sediment in g cm$^{-2}$ ka$^{-1}$ [53].

Extracting the dust component from the bulk sediment is done by calculating the $^{232}Th$ flux that is purely of continental origin[46] and assuming a constant concentration of Th in the regional dust[37]. The $^{232}Th$ flux is given by $PF_i = PF_T x f_i$, where $PF_i$ is the $^{232}Th$ flux and $f_i$ is the $^{232}Th$ concentration in the sample. Dividing the sample Th flux by the Th concentration in the regional dust results with the terrigenous deposition fluxes of the sample that is the dust accumulation rates.

A measure of the lateral sediment redistribution on the sea floor is expressed by the focusing factor ($\Psi$), which is calculated by dividing the $^{230}Th_{xs}^0$ found in the sediment by the production of $^{230}Th$ from the water column:

$$\Psi = \frac{\int_{h2}^{h1} \left( ^{230}Th_{xs}^0 \rho dh \right)}{Production_{230}(t_2 - t_1)} \quad (2)$$

Where $h$ denotes the depths in the core and $t$ the corresponding age, thus the limitation of this method is that it can be done solely on dated horizons[53,58]. The production is calculated with the appropriate water depth and $\rho$ is the density of the sediment. When $\Psi < 1$, it suggests that sediments carrying Th nuclides had been either advected prior to deposition or winnowed post deposition by bottom currents. Results where $\Psi > 1$ suggest the opposite process. However, values of $\Psi$ in all three studied cores (Supplementary Fig. 2) remain close to 1, suggesting that processes of sediment winnowing and focusing are negligible during the studied intervals.

## Data availability
The authors declare that the data supporting the findings of this study are available within the paper and its supplementary information files.

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

## Acknowledgements

Ahuva Almogi-Labin (Geological Survey of Israel) and C. Hemleben and H. Schulz (University of Tuebingen, Germany) are thanked for sharing samples and knowledge. We thank Ortal Sava for assisting with laboratory procedures. Some analyses and visualizations used in this paper were produced using the Giovanni online data system, developed, and maintained by the NASA GES DISC. Funding was provided by Israel Science Foundation grant 927/15 to A. Torfstein.

## Author contribution

A.T. designed the study and D.P. preformed the Th isotopes analyses. D.P. and A.T. jointly interpreted the results and wrote the manuscript.

## Additional information

**Competing interests:** The authors declare no competing interests.

