## [Peer Review File · Nature Communications]

Reviewers' comments:

Reviewer #1 (Remarks to the Author):

Thank you for your invitation to review the manuscript entitled [Sahara dust fluxes record the northern limits of the African Humid Period « green Sahara »] from Palchan and Torfstein.

This manuscript, which I read with great interest, contributes to document the last African Humid Period (AHP) characterized by major hydrological changes in Northern Africa around 12-5kyrs BP. This period has drawn considerable interest over the last decade or so, as it is one of the most prominent regional climate change that took place in the current Holocene interglacial epoch. Some earlier studies have suggested rapid, non-linear onset and termination of the AHP, likely involving positive feedback processes.

These findings have implications for our understanding of the climate dynamics in the region and its sensitivity to various forcings, and are therefore of particular interest for climate projections in this vulnerable region in the context of global warming.

The variability in the amount of mineral dust blown from the arid and semi-arid regions of the Sahara/Sahel region has proved to be particularly helpful to document switches in African climate between arid and wetter periods.

The most useful records so far have been obtained from marine sediments, off West, North and East Africa, in the Atlantic, the Mediterranean and the Gulf of Aden, respectively.

Changes in the mineral dust fluxes in marine sediments off West Africa have been most extensively documented, providing a relatively detailed picture of northern expansion of the AHP and its chronology on the western side of the Sahara.

By contrast, the north-eastern extent of the "Green Sahara" is less well documented yet, as it is largely based on continental records whose interpretation is often difficult and which are generally discontinuous.

The manuscript of Palchan and Torfstein provides, for the first time, a transect of three records of dust deposition flux obtained across the Red Sea.

These records are based on the 230Thxs measurements, which provide the most reliable sedimentation rate estimates and thus the best possible quantitative assessment of the dust fluxes.

These new data from the Red Sea show a limited reduction in dust fluxes (50%) compared to West Africa where it reached 80%, and, most importantly, a rather limited geographical extent of the AHP in East Africa.

This manuscript therefore enable to fill a gap in our knowledge of the AHP and also provides a useful review of earlier AHP studies, which are used by the author to redraw the global picture of the AHP expansion from West to East.

Overall, I find that the manuscript from Palchan and Torfstein is based on sound data, that it is clearly written, well organized and illustrated, and so I would therefore recommend publication Nature Communications nearly as is.

Below are some more detailed comments and minor modification suggestions that the author may want to consider before publication though.

Line 16 (and throughout the manuscript, third main paragraph starting line 51 for instance): a discussion on the timing of the AHP as shown by KL11 and KL15 is missing; even though the rather poor temporal resolution of the records (the main limitation of this study?) may not allow for a precise assessment of the duration of the AHP in the sector of Eastern Africa, this point should be mentioned and discussed (especially as the duration of the AHP is not identical in both records and that its termination appears earlier in KL15).

Similarly, the data points should be indicated on Fig. 2a (not only the envelopes) in order to show the temporal resolution of the records without having to go to the SI.

Line 42: I would replace by "230Thxs-normalized "dust" accumulation rates" (as it is what matters in the story)

figure 1 (first mentioned line 43): I find somewhat misleading the fact that the latitudinal range is

different on fig 1a and 1b: I think it should be identical both in the upper and lower panel. Same for the detrital flux colour scale. Also, it is unclear from the legend what the horizontal (solid and dash) and vertical lines indicate on the lower panels stand for.

Lines 45-46: the sentence reads funny, rephrase?

Line 55 and Fig 2b: why choosing the insolation at 30°N, and not at 20°N for instance?

Line 63 (and Fig SI): the assumption the authors seem to make here is that we can infer wind patterns across the entire Holocene from current wind patterns; this should be discussed or at least acknowledged in the text.

Line 84: Fig. 2d (not 2e)

Line 92: where is "the approximate 5 kyr of the AHP" coming from? earlier studies? KL 11 and 15 records? (again, a discussion on the duration and timing of onset/termination as showed by dust fluxes in the Red Sea is required).

Line 101: I could not find Chott lake and Ahmet Lake on Fig. 1.

Line 104: Fig 3 (not Fig 1)

Line 113-114: "between 10 and 5 ka": this information should come earlier (see my comments above)

Figure 2's legend: d is e, and vice versa

Figure 3: I am not convinced the main arrow indicating westward dust transport toward the Atlantic does reflect the back trajectories of Fig SI faithfully, as the latter often display a north-easterly component.

SI, lines 69-70: it is not clear why the different temporal resolution of the two records (KL15 dust flux or δD_{wax}) is a problem and why comparing the δD_{wax} of RC09 with KL11 is more suitable; please clarify.

Reviewer #2 (Remarks to the Author):

The manuscript main outcome is a proposed geographical limit to the northward extension of the green Sahara, in the context of the African Humid Period that characterized North African climate during the Early to mid-Holocene. The conclusions are reached based on the analysis of sedimentary cores from the same research group drilled in the Arabian Sea and Gulf of Aden, as well as from a critical review of the literature, notably in terms of proxy data for paleo-environmental conditions in the region. It appears that the original data contribution from this paper relates to core KL15 from the Gulf of Aden, whereas two other records from the Arabian Sea were presented in previous contributions from the same group, although additional analyses are also presented here.

Concerning the sediment cores from the Arabian Sea and Gulf of Aden, in this manuscript the passage is done quickly to translate "terrigenous component" into "dust", mainly based on previous results on grain size distributions and geochemical fingerprinting from radiogenic isotopes. This passage may warrant more attention if the scope was to provide quantitative estimates of the dust flux itself, but in the context of using this parameter as a proxy for aridity I think the approach can be considered acceptable.

On the other hand very little information is shown either here or in Palchan et al. 2013 about the age models of the three sediment cores. I think this should be addressed more extensively in the

supplement.

Also, there is no discussion on the degree of sediment focusing in the records.

Concerning the datasets from the literature, I found there is little justification provided for the selective consideration given (or not) to specific data.

For instance, the authors themselves mention their reconstruction "is in contrast with recent suggestions (8,11)" (line 274). In the case of (8: Tierney et al., 2017) the reference appears to point to the NW limit of the "green Sahara" extension, the difference being related to the interpretation of proxies from sediment core GC27. The authors in this manuscript refer to the dust record of GC27 (McGee et al., 2013), stating no large changes in fluxes are seen to justify their suggestion that the limit should be placed South of this core location, essentially. On the other hand Tierney et al. focus on the δD -wax data to infer an increase in precipitation as far as ~31 degrees North in NW Sahara. In this case, the δD -wax data from GC27 is disregarded, but the same proxy is highlighted in the case of the Gulf of Aden.

In addition, it is frankly surprising not to see any mention to yet other datasets, e.g. Bartlein et al. 2011. More in general, I think that the authors should put more emphasis on clearly explaining how they constructed their reference dataset.

In general, I find that this manuscript is potentially an interesting contribution to the scientific debate on the extension of the "green Sahara", which is an important topic that has received attention also in high-ranking journals. Nonetheless, I think that there are a few aspects that need to be addressed in more detail in order to establish how strong are the constraints that the authors use to justify their conclusions.

References

Palchan, D., Stein, M., Almogi-Labin, A., Erel, Y. & Goldstein, S. L. Dust transport and synoptic conditions over the Sahara–Arabia deserts during the MIS6/5 and 2/1 transitions from grain-size, chemical and isotopic properties of Red Sea cores. *Earth Planet. Sci. Lett.* 382, 125–139 (2013).

Tierney, J. E., Pausata, F. S. . & deMenocal, P. B. Rainfall regimes of the Green Sahara. *Sci. Adv.* 3, (2017).

McGee, D., deMenocal, P. B., Winckler, G., Stuut, J. B. W. & Bradtmiller, L. I. The magnitude, timing and abruptness of changes in North African dust deposition over the last 20,000 yr. *Earth Planet. Sci. Lett.* 371–372, 163–176 (2013).

Bartlein, P. J., et al. (2011), Pollen-based continental climate reconstructions at 6 and 21 ka: A global synthesis, *Clim. Dyn.*, 37(3–4), 775–802, doi:10.1007/s00382-010-0904-1.

Reviewer #3 (Remarks to the Author):

The authors of the paper "Sahara dust fluxes record the northern limits of the African Humid period "green Sahara".

The paper is presenting some synthesis about past 20 ka record of dust deposition in North Atlantic along the western coast between 19°N and 28°N and new data from 3 cores from the Red Sea and Gulf of Aden, an area from which few data are available to compare with. In the abstract, the authors claim that there are sporadic terrestrial records in the northeast Sahara and that it is more useful to deal with marine cores. This wording is problematic as although one could understand that marine cores could be interpreted as continuous, which are not always true, terrestrial records, even sporadic are evidences for the time concerned. Amazingly, later in the manuscript, the authors used these sporadic records to state about the northern limit they indicate on figure 3. Remaining on the sporadic terrestrial record, therefore what the authors are addressing is rather a latitudinal span on both side of the African continent than a more geographical one.

In the second paragraph, the authors give a general state of the art about the question addressed. However they refer to the occurrence of plants in a way, which is not acceptable. Indeed they indicate line 33-35: "Furthermore, vegetation reconstructions include individual migrating plants, which do not necessarily reflect a regional climate change but rather an incisional migration of some plants through sporadic waterways. " A first question is: "what is an incisional migration? More important, the authors rely on pollen results apparently. First of all, pollen is a very volatile material, which can be transported over extremely long distance. However, if particular plant adapted to humid conditions, the study the authors refer to speak about the migration of tropical plants, are migrating this may be due to more soil water availability and temperature. Therefore if there is more water in the soils, how does this happen without more precipitation, and through some climate change? Later in the same paragraph, the authors claim that they are addressing "Robust evidence of the environmental change" through marine cores. How more robust are their records?

In the forth paragraph, the authors are relating to values lines 51-53. However there are envelopes, which would be better to refer to rather than using the average value. Furthermore in central Red Sea, HS-1 shows some variation and the dust flux is even higher after HS-1, a feature not explained or discussed in the manuscript. In several places in the manuscript, the authors refer to data which are not shown making the reading of the text difficult and leading the reader to question the reliability of the argument. Line 59, the authors refer to "depleted Mediterranean foraminifera $\delta^{18}O$ values. But there is a strong variability although the average value remains stable through the past 12 ka.

On figure 1, tick for latitude and longitude are missing. The caption does not explain what AOD means nor what is the unit while this information are given in the caption of figure 3? Moreover there is a mistake in the detrital flux unit on the right hand lower panel.

On figure 2, it's a pity that the mode variations prior 15 ka so before HS-1, are not available because they could have been good arguments in the discussion

On figure 3 the authors, line 252, refer to Giovanni online data but with no reference. Same comment about the mega lake "Chad, the wetter conditions during the AHP. Also line s 261-262, the authors refer to various records, given the bibliographical reference but without plotting the sites on the map.

The general appreciation of the reading of this manuscript is that it is not yet complete, the data obtained being not enough digested and discussed properly. Even the manuscript keeps some errors that should not remain when submitting a paper. I strongly recommend the authors to reconsider their manuscript, better develop their arguments and submit the revised paper in a more specialized journal.

denis-didier Rousseau

We hereby provide our response to the comments made by three reviewers on an earlier version of this manuscript considered by *Nature communications*. Our response is highlighted in green.

Reviewers' comments:

Reviewer #1 (Remarks to the Author):

Thank you for your invitation to review the manuscript entitled [Sahara dust fluxes record the northern limits of the African Humid Period « green Sahara »] from Palchan and Torfstein.

This manuscript, which I read with great interest, contributes to document the last African Humid Period (AHP) characterized by major hydrological changes in Northern Africa around 12-5kyrs BP.

This period has drawn considerable interest over the last decade or so, as it is one of the most prominent regional climate change that took place in the current Holocene interglacial epoch.

Some earlier studies have suggested rapid, non-linear onset and termination of the AHP, likely involving positive feedback processes.

These findings have implications for our understanding of the climate dynamics in the region and its sensitivity to various forcings, and are therefore of particular interest for climate projections in this vulnerable region in the context of global warming.

The variability in the amount of mineral dust blown from the arid and semi-arid regions of the Sahara/Sahel region has proved to be particularly helpful to document switches in African climate between arid and wetter periods.

The most useful records so far have been obtained from marine sediments, off West, North and East Africa, in the Atlantic, the Mediterranean and the Gulf of Aden, respectively.

Changes in the mineral dust fluxes in marine sediments off West Africa have been most extensively documented, providing a relatively detailed picture of northern expansion of the AHP and its chronology on the western side of the Sahara.

By contrast, the north-eastern extent of the “Green Sahara” is less well documented yet, as it is largely based on continental records whose interpretation is often difficult and which are generally discontinuous.

The manuscript of Palchan and Torfstein provides, for the first time, a transect of three records of dust deposition flux obtained across the Red Sea.

These records are based on the $^{230}\text{Th}_{\text{xs}}$ measurements, which provide the most reliable sedimentation rate estimates and thus the best possible quantitative assessment of the dust fluxes.

These new data from the Red Sea show a limited reduction in dust fluxes (50%)

compared to West Africa where it reached 80%, and, most importantly, a rather limited geographical extent of the AHP in East Africa.

This manuscript therefore enables to fill a gap in our knowledge of the AHP and also provides a useful review of earlier AHP studies, which are used by the author to redraw the global picture of the AHP expansion from West to East.

Overall, I find that the manuscript from Palchan and Torfstein is based on sound data, that it is clearly written, well organized and illustrated, and so I would therefore recommend publication in *Nature Communications* nearly as is.

Below are some more detailed comments and minor modification suggestions that the author may want to consider before publication though.

1. Line 16 (and throughout the manuscript, third main paragraph starting line 51 for instance): a discussion on the timing of the AHP as shown by KL11 and KL15 is missing; even though the rather poor temporal resolution of the records (the main limitation of this study?) may not allow for a precise assessment of the duration of the AHP in the sector of Eastern Africa, this point should be mentioned and discussed (especially as the duration of the AHP is not identical in both records and that its termination appears earlier in KL15).

We thank the reviewer for this comment. The available constraints on the timing of the AHP are such that it is not possible at this point to identify leads or lags between different regions. Alternatively, the main point that is made in this context, is that the AHP-related “wetness” did not reach north of 22°N during the AHP. We readily agree that better chronological constraints are required to investigate the sub-millennial temporal dynamics during this period, but obtaining these is beyond the scope of the current discussion.

2. Similarly, the data points should be indicated on Fig. 2a (not only the envelopes) in order to show the temporal resolution of the records without having to go to the SI.

We agree, though the temporal resolution can be clearly (and more easily) appreciated in Figure 1 (bottom panels), while adding the actual data points to figure 2 has proven to be less elegant. We thus leave this graphic issue unchanged.

#See changes in Fig. 2

3. Line 42: I would replace by “230Thxs-normalized “dust” accumulation rates” (as it is what matters in the story)

Agreed. We changed the subject in the relevant places from “sediment” to “dust”.

4. figure 1 (first mentioned line 43): I find somewhat misleading the fact that the latitudinal range is different on fig 1a and 1b: I think it should be identical both in the upper and lower panel.

The upper panels show the geographical location of the cores, and hence, require a somewhat extended window so the reader is able to easily identify the location of the cores. The lower panels are based on the data with some interpolation; however, extending the windows of the lower panels will result with extrapolation that could be misleading. We did however mark the site locations more clearly.

5. Same for the detrital flux colour scale.

We thank the reviewer for this comment. We do however note that there is no reason to assume that fluxes are symmetrical in both sectors of the Sahara Desert. Moreover, it is clear that there could be expected to be a connection between the flux value and the distance of the study site from the source. Thus, using the same scale results in a strong smearing of the signal (in the Red Sea sector), which we argue is irrelevant when the focus is on the spatiotemporal patterns of dust fluxes in the region. Using regionally-tuned scales allows a much better evaluation of these patterns.

6. Also, it is unclear from the legend what the horizontal (solid and dash) and vertical lines indicate on the lower panels stand for.

We agree, and hence we removed them.

7. Lines 45-46: the sentence reads funny, rephrase?

OK. We rephrased to clearly state why the detritus in the Red Sea cores mainly reflects dust and not fluvial contributions.

Lines 90-92.

8. Line 55 and Fig 2b: why choosing the insolation at 30°N, and not at 20°N for instance?

We accept this comment. In the revised version we now use the “monsoon index” that is the sum of the insolation at the tropic of cancer and the gradient between insolation at the tropic of cancer and at the equator (Rossignol-Strick, 1985). Indeed, the observed environmental changes display the best correlation with the monsoon index (rather than the semi-arbitrarily chosen insolation curves).

#Line 111 and Fig. 2b.

9. Line 63 (and Fig SI): the assumption the authors seem to make here is that we can infer wind patterns across the entire Holocene from current wind patterns; this should be discussed or at least acknowledged in the text.

This is an important point and we thank the reviewer for bringing it up. Indeed, this point is debated in the SI and we address it now also in the main body of the manuscript and in the figure caption.

#Lines 92-99.

10. Line 84: Fig. 2d (not 2e)

Corrected.

11. Line 92: where is “the approximate 5 kyr of the AHP” coming from? earlier studies? KL 11 and 15 records? (again, a discussion on the duration and timing of onset/termination as showed by dust fluxes in the Red Sea is required).

Constraints on the duration of the AHP in western Africa, as recorded by dust deposited in the Atlantic Ocean are mostly derived from deMenocal et al. (2000). More recent studies on dust accumulation rates show similar ages for the reduced western Saharan dust plume (McGee et al., 2013). We rephrased the sentence to be more clear and added the reference. Also, see our response to comment #1 about this issue.

#Lines 42-44.

12. Line 101: I could not find Chott lake and Ahmet Lake on Fig. 1.

We corrected the text accordingly (Fig. 1 changed to Fig. 3).

Line 104: Fig 3 (not Fig 1)

We corrected the text accordingly (Fig. 1 changed to Fig. 3).

13. Line 113-114: “between 10 and 5 ka”: this information should come earlier (see my comments above)

We agree. Indeed, in the revised manuscript this issue is discussed earlier when describing the eastern Saharan dust plume evolution during the deglaciation and the Holocene: *“Through the last deglaciation, dust accumulation rates in the Gulf of Aden and the central Red Sea dropped from maximum values of $0.72 \text{ g cm}^{-2} \text{ ka}^{-1}$ and $0.89 \text{ g cm}^{-2} \text{ ka}^{-1}$ during the deglacial to minimum values of $0.40 \text{ g cm}^{-2} \text{ ka}^{-1}$ and $0.35 \text{ g cm}^{-2} \text{ ka}^{-1}$ at ca. 7 ka, respectively (Fig. 1 & Fig. 2a)”*. #Lines 104-106.

The timing of the AHP is now also described in the abstract. #Lines 12-13.

14. Figure 2's legend: d is e, and vice versa

Fixed.

15. Figure 3: I am not convinced the main arrow indicating westward dust transport toward the Atlantic does reflect the back trajectories of Fig SI faithfully, as the latter often display a north-easterly component.

The revised version of the figure has been modified so that the arrow reflects the northeasterly component as well.

16. SI, lines 69-70: it is not clear why the different temporal resolution of the two records (KL15 dust flux or δD_{wax}) is a problem and why comparing the δD_{wax} of RC09 with KL11 is more suitable; please clarify.

The problem stems from comparing two different cores with samples that are not necessarily coeval. In the revised version we created an interpolated data set which was used to compare data from core KL11 and core KL15 on the same timescale.

#Line 121-124 (and in the supplementary section 2).

Reviewer #2 (Remarks to the Author):

The manuscript main outcome is a proposed geographical limit to the northward extension of the green Sahara, in the context of the African Humid Period that characterized North African climate during the Early to mid-Holocene. The conclusions are reached based on the analysis of sedimentary cores from the same research group drilled in the Arabian Sea and Gulf of Aden, as well as from a critical review of the literature, notably in terms of proxy data for paleo-environmental conditions in the region.

17. It appears that the original data contribution from this paper relates to core KL15 from the Gulf of Aden, whereas two other records from the Arabian Sea were presented in previous contributions from the same group, although additional analyses are also presented here.

This comment is quite incorrect. The data set that triggered the discussion reflected in this manuscript are the $^{230}\text{Thxs}$ -normalized sediment (dust) accumulation rates in the Red Sea and Gulf of Aden. This data is published here for the first time, and in fact is the first application of $^{230}\text{Thxs}$ as a constant flux proxy in this region. Naturally, in order to frame the discussion in the context of a large picture setting, we combined the results with corresponding observations (of $^{230}\text{Thxs}$) from West Africa and the Arabian Sea, as well as previous records of geochemical and sedimentological proxies from the Red Sea (e.g., Fig. 2). We further expanded the discussion to a comprehensive compilation of relevant observations across the Sahara Desert (e.g., Fig. 3).

Thus, we do indeed use previous observations to discuss the results, but this manuscript includes two very new data sets: 1. $^{230}\text{Thxs}$ -normalized sediment (dust) accumulation rates in the Red Sea and Gulf of Aden, 2. Compilation of environmental-climatic records from marine and continental archives across and surrounding the Sahara Desert during the AHP.

Combined, our conclusions are new, providing for the first time absolute dust accumulation rates in the eastern Sahara Desert and their climatic context, and more

importantly, they regional compilation reconciles for the first time between ample evidence reported in the literature that pertains to the AHP. Thus the suggestion that the data discussed here has already been published is not correct.

18. Concerning the sediment cores from the Arabian Sea and Gulf of Aden, in this manuscript the passage is done quickly to translate “terrigenous component” into “dust”, mainly based on previous results on grain size distributions and geochemical fingerprinting from radiogenic isotopes. This passage may warrant more attention if the scope was to provide quantitative estimates of the dust flux itself, but in the context of using this parameter as a proxy for aridity I think the approach can be considered acceptable.

We thank the reviewer for this comment and would like to clarify. Converting the terrigenous accumulation rates to dust accumulation rates is done using several arguments. First, it must be considered that the regional hyperaridity together with the very small catchments along the Red Sea, limit the direct fluvial contributions to be negligible: *“Because the drainage basins that surround the Red Sea are relatively small, and because of the hyperarid conditions that limit direct fluvial contributions to be negligible relative to the desert dust plumes³¹, the terrigenous fraction in the Red Sea bottom sediments is overwhelmingly of eolian origin”*, #Lines 90-92. and by the isotope composition of the detritus *“The provenance of the terrigenous fraction of downcore records in the Red Sea, based on their radiogenic isotope composition, confirms that the latter dust sources were similarly active during the late Quaternary and the Holocene”*. #Lines 97-99. Moreover, the correlation of the dust MAR to hematite concentrations further supports these assumptions, as it was shown that hematite concentrations reflect Saharan dust (Larrasoana et al., 2008; Roberts et al., 2011). #Line 127-129.

19. On the other hand very little information is shown either here or in Palchan et al. 2013 about the age models of the three sediment cores. I think this should be addressed more extensively in the supplement.

We agree. In the revised version the age model is provided in the supplementary information. Nevertheless, we use existing age models (Gieselhart, 1998; Grant et al., 2012, 2014; Hartman et al., in revision). The age modifications by Hartmann et al. are minor and no impact on the discussion or the conclusions. Regardless, discussing the development of the age models is beyond the scope of the current paper.

20. Also, there is no discussion on the degree of sediment focusing in the records.

This is a very important comment. We readily acknowledge that discussion of sediment focusing was not adequately addressed in the original version. In the revised

manuscript, we discuss the focusing factors in the studied cores and provide more details in the supplementary information:

“All three cores show stable and close to 1 focusing factors suggesting the dust fluxes reflect mostly vertical deposition with minor sediment focusing or winnowing (Supplementary section 1).” #Line 113-115.

21. Concerning the datasets from the literature, I found there is little justification provided for the selective consideration given (or not) to specific data. For instance, the authors themselves mention their reconstruction “is in contrast with recent suggestions (8,11)” (line 274). In the case of (8: Tierney et al., 2017) the reference appears to point to the NW limit of the “green Sahara” extension, the difference being related to the interpretation of proxies from sediment core GC27. The authors in this manuscript refer to the dust record of GC27 (McGee et al., 2013), stating no large changes in fluxes are seen to justify their suggestion that the limit should be placed South of this core location, essentially. On the other hand Tierney et al. focus on the δD -wax data to infer an increase in precipitation as far as ~31 degrees North in NW Sahara. In this case, the δD -wax data from GC27 is disregarded, but the same proxy is highlighted in the case of the Gulf of Aden.

Indeed, at core GC27 the dust accumulation record, which averages environmental conditions over large geographic extent, shows different timely evolution when compared with the other cores presented in the same study (McGee et al., 2013). We interpret this as indicating that the AHP did not affect the dust accumulation rates (and hence, its emissions) in the northernmost reach of the Sahara Desert. Yet, the δD -wax record does record relatively wetter conditions at an interval that partly overlaps with the AHP (Tierney et al., 2017). We address this discrepancy and argue that the δD -wax reflects the conditions at the Middle and High Atlas Mts. (exceeding 2000 m elevation) and not the vast northern Sahara Desert. Furthermore, there seems to be a northern source for increased precipitation in the record presented in Tierney’s work, which was previously postulated to be from Atlantic cyclones (Wassenburg et al., 2016). We address this issue in detail in the revised text. #Line 228-232.

22. In addition, it is frankly surprising not to see any mention to yet other datasets, e.g. *Bartlein et al. 2011*. More in general, I think that the authors should put more emphasis on clearly explaining how they constructed their reference dataset.

We thank the reviewer for his suggestion to incorporate this global synthesis; indeed the conclusions made by Bartlein et al., 2011 corroborate our suggestion for the AHP environmental change. *“These latitudes are also supported by pollen based reconstructions of soil humidity and precipitation⁴⁷.”* #Lines 217-218.

23. In general, I find that this manuscript is potentially an interesting contribution to the scientific debate on the extension of the “green Sahara”, which is an important topic that has received attention also in high-ranking journals. Nonetheless, I think that there are a few aspects that need to be addressed in more detail in order to establish how strong are the constraints that the authors use to justify their conclusions.

We readily agree regarding the importance and interest of the community in the AHP and climate patterns across the Sahara Desert. Clearly there are still many knowledge gaps that require consideration but this manuscript attempts to tie together observations and records from a vast region, in order to deconvolve the spatiotemporal impact of the AHP.

References

Palchan, D., Stein, M., Almogi-Labin, A., Erel, Y. & Goldstein, S. L. Dust transport and synoptic conditions over the Sahara–Arabia deserts during the MIS6/5 and 2/1 transitions from grain-size, chemical and isotopic properties of Red Sea cores. *Earth Planet. Sci. Lett.* 382, 125–139 (2013).

Tierney, J. E., Pausata, F. S. . & deMenocal, P. B. Rainfall regimes of the Green Sahara. *Sci. Adv.* 3, (2017).

McGee, D., deMenocal, P. B., Winckler, G., Stuut, J. B. W. & Bradtmiller, L. I. The magnitude, timing and abruptness of changes in North African dust deposition over the last 20,000 yr. *Earth Planet. Sci. Lett.* 371–372, 163–176 (2013).

Bartlein, P. J., et al. (2011), Pollen-based continental climate reconstructions at 6 and 21 ka: A global synthesis, *Clim. Dyn.*, 37(3–4), 775–802, doi:10.1007/s00382-010-0904-1.

Reviewer #3 (Remarks to the Author):

The authors of the paper “Sahara dust fluxes record the northern limits of the African Humid period “green Sahara”.

The paper is presenting some synthesis about past 20 ka record of dust deposition in North Atlantic along the western coast between 19°N and 28°N and new data from 3 cores from the Red Sea and Gulf of Aden, an area from which few data are available to compare with.

24. In the abstract, the authors claim that there are sporadic terrestrial records in the northeast Sahara and that it is more useful to deal with marine cores. This wording is problematic as although one could understand that marine cores could be interpreted as continuous, which are not always true, terrestrial records, even sporadic are evidences for the time concerned. Amazingly, later in the manuscript, the authors used these

sporadic records to state about the northern limit they indicate on figure 3. Remaining on the sporadic terrestrial record, therefore what the authors are addressing is rather a latitudinal span on both side of the African continent than a more geographical one.

We agree, and changed the abstract accordingly; we now focus on the debate regarding the latitudinal extent of the monsoonal precipitation during the AHP and not on the comparison of terrestrial and marine records. #Lines 4-20.

25. In the second paragraph, the authors give a general state of the art about the question addressed. However they refer to the occurrence of plants in a way, which is not acceptable. Indeed they indicate line 33-35: "Furthermore, vegetation reconstructions include individual migrating plants, which do not necessarily reflect a regional climate change but rather an incisional migration of some plants through sporadic waterways. " A first question is: "what is an incisional migration? More important, the authors rely on pollen results apparently. First of all, pollen is a very volatile material, which can be transported over extremely long distance. However, if particular plant adapted to humid conditions, the study the authors refer to speak about the migration of tropical plants, are migrating this may be due to more soil water availability and temperature. Therefore if there is more water in the soils, how does this happen without more precipitation, and through some climate change?

It is important to clarify that we do not argue against the climate change that took place in northern Africa during the early Holocene, on the contrary, this is stated in the manuscript several times, and supported by the dust fluxes we present. However, the Sahara Desert requires more attention to the details. Adaptation of the tropical plants, indeed as documented in pollen records, in the Sahara Desert is found in riparian sites (Hély et al., 2014; Watrin et al., 2009). This strongly suggests that it is not a regional climate change at the Sahara Desert, but rather, a localized transformation (due to increased soil moisture and water availability) that occurred on river banks and in the proximity of fresh waterbodies- that is incisional (as opposed to regional), we changed the text accordingly to be clearer. #Line 33-36.

26. Later in the same paragraph, the authors claim that they are addressing "Robust evidence of the environmental change" through marine cores. How more robust are their records?

Here, "robust" refers to the nature of analysis that is quantitative compared with pollen counting at sporadic sites. Moreover, the generally continuous nature of marine and lacustrine cores allows the reconstruction of continuous records in arid regions that are often subject to hiatuses either from erosion or lack of a "trap". In light of the reviewers comment, we did however revise the text in order to express more clearly the above point.

Lines 36-39 and lines 41-42.

27. In the forth paragraph, the authors are relating to values lines 51-53. However there are envelopes, which would be better to refer to rather than using the average value.

The data and related uncertainties are provided in the table (and in the figures). The coherence and readability are better served by reporting the average values, and this does not alter the bottom line. Hence, we decided to report the average values in the text and rely on the data tables for the report of the full range of data.

28. Furthermore in central Red Sea, HS-1 shows some variation and the dust flux is even higher after HS-1, a feature not explained or discussed in the manuscript.

We agree and thank the reviewer for this comment. In the revised version we addressed this point in the captions of figure 1. #Lines 85-88.

29. In several places in the manuscript, the authors refer to data which are not shown making the reading of the text difficult and leading the reader to question the reliability of the argument.

We are not sure what the reviewer is referring to here. Yet we made great efforts to ensure that all the relevant data discussed in the revised version is clearly presented, including appropriate references.

30. Line 59, the authors refer to "depleted Mediterranean foraminifera $\delta^{18}\text{O}$ values. But there is a strong variability although the average value remains stable through the past 12 ka.

Indeed, the $\delta^{18}\text{O}$ data displays variability that may be confusing. Thus, we use in the new version a 5-point smoothed curve which provides a much more clear picture of the pattern of $\delta^{18}\text{O}$ change across the early Holocene.

31. On figure 1, tick for latitude and longitude are missing.

Corrected

32. The caption does not explain what AOD means nor what is the unit while this information are given in the caption of figure 3? Moreover there is a mistake in the detrital flux unit on the right hand lower panel.

Fixed. #Lines 78-80.

33. On figure 2, it's a pity that the mode variations prior 15 ka so before HS-1, are not available because they could have been good arguments in the discussion

Indeed, we couldn't agree more. However, this data is not available, and in fact the data presented in this diagram is shown for the first time, almost 30 years after the cores were extracted, and we can only hope that this will encourage further research in this region.

34. On figure 3 the authors, line 252, refer to Giovanni online data but with no reference.

We corrected this. The reference was added to the acknowledgments according to the Giovanni site instructions. #Lines 78-80

35. Same comment about the mega lake "Chad, the wetter conditions during the AHP.

We thank the reviewer to note this point, we have fixed this issue and added the reference to the appropriate place in the captions "*and a blue square indicates the approximate location of "mega lake" Chad¹⁰*". #Lines 204-205.

36. Also lines 261-262, the authors refer to various records, given the bibliographical reference but without plotting the sites on the map.

We have rephrased the sentence and added the appropriate citations. All mentioned records are plotted in the map. For further investigation the readers can follow the references. "The green line was drawn based on previous compilations from: lacustrine and palustrine deposits^{8,15,45}, vegetation based modeling⁴⁶, leaf wax data⁷, mineralogical studies in the Mediterranean¹⁶ and the suggested "mega lakes"¹⁰." #Lines 208-211.

37. The general appreciation of the reading of this manuscript is that it is not yet complete, the data obtained being not enough digested and discussed properly. Even the manuscript keeps some errors that should not remain when submitting a paper. I strongly recommend the authors to reconsider their manuscript, better develop their arguments and submit the revised paper in a more specialized journal.

denis-didier Rousseau

We thank the reviewer for this comment and for highlighting some weak points in the manuscript. The revised version includes a larger data compilation and a more developed discussion. We made great efforts to encompass as many relevant references as permitted by the nature-group submission policy, and further polished and honed the text.

We believe the newly submitted version provides a comprehensive discussion of cross-Sahara records of the AHP period, which has not been previously presented in this way. The manuscript also reports for the very first time the absolute sediment and dust accumulation rates in the eastern Sahara Desert sector, and shows that model predictions are about an order of magnitude offset to observations, warranting a comprehensive re-tuning of models. Given the important new insights detailed here and throughout the manuscript, we believe the contribution of this work is significant, spans several disciplines, and will have a long lasting

impact on studies of climate change and dust patterns in the Sahara Desert region over the later Quaternary.

Reviewers' comments:

Reviewer #1 (Remarks to the Author):

As stated in my initial review, I find that Palchan and Torfstein's manuscript fills a important gap in our knowledge of the African Humid Period by providing, for the first time, dust flux data from the Red Sea, i.e. the Eastern edge of the Saharan Desert. These data, combined with evidence obtained earlier across the Sahara region, in the Mediterranean, and off West Africa, enable the author to redraw the AHP northward expansion from West to East. So, I find that the manuscript provides a useful overview of the AHP, which should be helpful for a wide range of scientists, including climate modelers. I therefore have no doubt this paper will be highly cited.

Overall, I am satisfied with the author's answers to my comments and I find that the discussion of the various data taken from literature in particular has improved in the revised version of the manuscript, strengthening the authors' main claims.

I would therefore reiterate my initial recommendation to accept the manuscript for publication in Nature Communications.

Two minor comments:

Line 91: it seems to me as if some words are missing in this sentence

Figure 2: if, as stated in the legend, the monsoon index represents the difference in insolation between the tropic of Cancer and the equator, it looks as if the values indicated on the y-axis of Fig 2b are very high !? (also, could the author specify which insolation is used in the calculation? July? summer? yearly?)

Reviewer #2 (Remarks to the Author):

The new version of the manuscript addresses some of the points raised by the reviewers. Nonetheless it still contains some points to be clarified, and some errors. Those issues should be addressed properly before I can recommend that the manuscript is accepted for publication. I would encourage the authors to do so because their data could be very useful.

The two main contributions of this work, according to the authors themselves, are new data and a holistic review of the existing literature on the subject.

As for the new data, the key finding is the different temporal evolution of dust fluxes between core KL11 and the more northern KL23. It is therefore of paramount importance that there is a clear explanation of how the dust component is defined and isolated from other lithogenic contributions and how well the chronology and stratigraphy are defined. Some clarifications on these aspects have enriched the revised manuscript, but a few points still need better clarification in my opinion. In particular it is not yet clear to me how the grain size analyses impact the dust flux calculations. For instance, in lines 131-136 it is stated that local floods are associated to a fining in the grain size of core KL11. I do not understand if this fine material is supposedly derived from eolian deflation of fine sediments deposited on land following these flooding events, or rather if it is material derived from river discharge into the Red Sea. If the latter is the case, this has some implications for the terrigenous fluxes. Such a record would show important information per se, but if an additional scope is to quantify the eolian input, either for addressing its variation with respect to other cores or to serve as a constraint for models, this must be somehow quantified and corrected (e.g. as in Ref. 23). Please clarify these issues in relation to all the new data presented here.

A previous comment I had made was in relation to the data/variable selection criteria adopted for the assemblage of the reference dataset as depicted e.g. in Figure 2. The authors provided some explanations in their response and added a line suggesting that the pollen dataset by Bartlein et al. support their finding, without going into details. I would suggest to use some of the space in the supplement to report the responses so they are accessible to the readers, as well as to elaborate a bit more on how specific choices were made.

The revised version of the manuscript now include an expanded and separated section dedicated to comparing the dust flux data with model simulations. This is an important exercise, and I agree that the new data, once clarified a few aspects as detailed above, could be very useful in providing a "target" for models. Unfortunately, this section is contains a few imprecise statements and errors that hamper its usefulness as it is.

First, it is stated that models "consider the Sahara Desert as one region" (272). This seems to imply, by reading the following line, that there is somehow a constant dust source all over North Africa. This is incorrect. Basically all global climate models, including the ones cited here, simulate dust emissions as a function of wind speed and certain surface characteristics, discretized on a gridded surface with horizontal resolution around 1x1 or 2x2 degrees latitude by longitude.

Second, when comparing the data to the model it is written that models do not account for millennial-timescale variations on dust accumulation rates (275-276). This is certainly true by definition in case of simulations in equilibrium conditions for a given climate state, e.g. the mid-Holocene at 6000 years before present. A meaningful comparison with this kind of simulations should focus on a comparable period. Thus this sentence does not provide any useful information.

Third, it is said that the models simulations in question overestimate dust deposition in the Red Sea cores by an order of magnitude (e.g. Figure 4). If we compare the values after the appropriate unit conversion, we can see that this is not true.

Fourth, it is reported that models fail to capture significant differences between the late Holocene and the mid Holocene values (297-298), but it is not clear on which grounds this statement is based.

Finally, the extrapolation segment for the KL15 dust flux curve from Figure 1a should be removed.

Reviewer #3 (Remarks to the Author):

Dear authors,

After reading the revised version of the manuscript, I must acknowledge that the authors are considerably improved the initial version of their manuscript, following the reviewers recommendations and comments, clarifying the points which were raising questions. On my side, I appreciate the effort to compile most available data, noticing that by now the manuscript refers to 53 papers compare to the initial 30. I have carefully checked both version and feel much more happy with the present version than with the initial one and the addition of figure 4 also clarify the authors' conclusion.

In the manuscript remain some minor corrections to be made like line 38, misspelling of "environments"

line 223 and 234 you write "grotte de Pist" while on figure 3 it reads "Grotte de PistE". In Wassenburg et al paper it is "Grotte de Piste". So please correct in the text.

line 250, when referring to earlier interglacial peaks I would rather use the chronological arrow by writing "MIS9,7, 5e. Same comment line 297, rather writing "between the mid Holocene and the late Holocene.

Reviewers' comments:

Reviewer #1 (Remarks to the Author):

1. As stated in my initial review, I find that Palchan and Torfstein's manuscript fills a important gap in our knowledge of the African Humid Period by providing, for the first time, dust flux data from the Red Sea, i.e. the Eastern edge of the Saharan Desert. These data, combined with evidence obtained earlier across the Sahara region, in the Mediterranean, and off West Africa, enable the author to redraw the AHP northward expansion from West to East. So, I find that the manuscript provides a useful overview of the AHP, which should be helpful for a wide range of scientists, including climate modelers. I therefore have no doubt this paper will be highly cited.
Overall, I am satisfied with the author's answers to my comments and I find that the discussion of the various data taken from literature in particular has improved in the revised version of the manuscript, strengthening the authors' main claims. I would therefore reiterate my initial recommendation to accept the manuscript for publication in Nature Communications.
2. Two minor comments:
Line 91: it seems to me as if some words are missing in this sentence.
We thank the reviewer for this comment. This sentence describes the considerations supporting the assumption that the terrigenous material in the Red Sea is mainly comprised of eolian material. While there were no missing words, we readily accept that the phrasing could be improved, and indeed we did so:
“Considering that the drainage basins that surround the Red Sea are relatively small, and given the regional hyperarid conditions that limit direct fluvial contributions to be negligible relative to the desert dust plumes³¹, the terrigenous fraction in the Red Sea bottom sediments is considered to be overwhelmingly of eolian origin.”
3. Figure 2: if, as stated in the legend, the monsoon index represents the difference in insolation between the tropic of Cancer and the equator, it looks as if the values indicated on the y-axis of Fig 2b are very high !? (also, could the author specify which insolation is used in the calculation? July? summer? yearly?)
The Monsoon index is calculated following Rossignol, 1983:
$$\text{Index} = \text{Insol.}_{(C)} + (\text{Insol.}_{(C)} - \text{Insol.}_{(E)})$$

(C)- tropic of cancer, (E)- equator. Thus, the values are high. The insolation used for the calculation is for August. We thank the reviewer for pointing this out and corrected the captions accordingly to show the equation and describe the insolation values that were used.

Reviewer #2 (Remarks to the Author):

4. The new version of the manuscript addresses some of the points raised by the reviewers.
We addressed all the points made by the reviewers, and provided in this file details on the modifications or response to each of the comments.

5. Nonetheless it still contains some points to be clarified, and some errors. Those issues should be addressed properly before I can recommend that the manuscript is accepted for publication. I would encourage the authors to do so because their data could be very useful.

We thank the reviewer for his efforts to aid improving this paper. Indeed, we argue this is a unique data set that provides the first record of absolute dust fluxes in the Holocene across the Red Sea and Gulf of Aden, and will therefore be highly interesting and useful for a wide range of scientists and enthusiasts.

6. The two main contributions of this work, according to the authors themselves, are new data and a holistic review of the existing literature on the subject. As for the new data, the key finding is the different temporal evolution of dust fluxes between core KL11 and the more northern KL23. It is therefore of paramount importance that there is a clear explanation of how the dust component is defined and isolated from other lithogenic contributions.

We accept this comment and in the revised version made considerable efforts to provide a clear description of the approach and methods. After calculating the sediment fluxes (PF) using the $^{230}\text{Th}_{\text{xs}}$ method we use the ^{232}Th concentration to extract the lithogenic component. *“The ^{232}Th flux is given by $PF_i = PF_T * f_i$, where PF_i is the ^{232}Th flux and f_i is the ^{232}Th concentration in the sample.”* This is given in the supporting information file. As for the eolian origins of the terrigenous component- we base this interpretation on a crucial assumption which arises from several observations: (1) Due to the uplifted shoulders of the Red Sea – it is surrounded by young mountain ranges from both sides – the catchments of wadies (dry river beds) that drain to the Red Sea are small in size. This suggests that in order to produce floods in these wadies the precipitation must occur directly above it (i.e., not out of its catchment). (2) The region is extremely hyper arid (today), rendering fluvial floods extremely rare. Hence, riverine sediment is negligible within the Red Sea terrigenous sediments:

“Considering that the drainage basins that surround the Red Sea are relatively small, and given the regional hyperarid conditions that limit direct fluvial contributions to be negligible relative to the desert dust plumes³¹, the terrigenous fraction in the Red Sea bottom sediments is considered to be overwhelmingly of eolian origin”.

7. ... and how well the chronology and stratigraphy are defined. Some clarifications on these aspects have enriched the revised manuscript, but a few points still need better clarification in my opinion.

We broadened the relevant supplementary section and added a detailed description (section 1.1) of the lithology and oxygen stratigraphy of the cores, and their implications regarding the studied region.

8. In particular it is not yet clear to me how the grain size analyses impact the dust flux calculations. For instance, in lines 131-136 it is stated that local floods are associated to a fining in the grain size of core KL11. I do not understand if this fine material is

supposedly derived from eolian deflation of fine sediments deposited on land following these flooding events, or rather if it is material derived from river discharge into the Red Sea. If the latter is the case, this has some implications for the terrigenous fluxes. Such a record would show important information per se, but if an additional scope is to quantify the eolian input, either for addressing its variation with respect to other cores or to serve as a constraint for models, this must be somehow quantified and corrected (e.g. as in Ref. 23). Please clarify these issues in relation to all the new data presented here.

This is a very good point and we thank the reviewer for bringing it up because it emphasizes our conclusions, as hereafter explained.

To clarify, we claim that despite the statement that "...the terrigenous fraction in the Red Sea bottom sediments is considered to be overwhelmingly of eolian origin", an exception is observed at core KL11 during the AHP, when this site received an additional influx of fluvial sediments, as suggested from the finning of the non-carbonate grain size distributions. This exception is of small proportions relative to the eolian contribution, as the overall detritus mass accumulation rates drop during the AHP. Indeed, the implication of this combined observation is that the true deposition rate of eolian material during the AHP at site KL11 is even smaller than the *a-priori* values reported here, and therefore providing further support to the argument that regional dust fluxes declined during the AHP. Thus, the discrepancy between observations and model predictions is even larger than described, and although we cannot determine the magnitude of this "excess" discrepancy, it is clear that dust emissions eastward from Africa decreased by more than 50% as suggested from the current observations, and perhaps even by 80% as in the westward dust plume. Clarification of this issue will be performed in a future study, which will focus on a combined grain size and provenance study.

Finally, it is worth noting that the source of the excess fluvial material during the AHP is probably from the Baraka basin, which is one of the larger drainage basins in the Red Sea watershed.

9. A previous comment I had made was in relation to the data/variable selection criteria adopted for the assemblage of the reference dataset as depicted e.g. in Figure 2. The authors provided some explanations in their response and added a line suggesting that the pollen dataset by Bartlein et al. support their finding, without going into details. I would suggest to use some of the space in the supplement to report the responses so they are accessible to the readers, as well as to elaborate a bit more on how specific choices were made.

We thank the reviewer for this useful comment. Accordingly, we added a section to the supplementary file where we provide a detailed yet clear explanation describing the results and context of observations from each of the data sets we use in our spatial compilation.

10. The revised version of the manuscript now include an expanded and separated section dedicated to comparing the dust flux data with model simulations. This is an important exercise, and I agree that the new data, once clarified a few aspects as detailed above, could be very useful in providing a "target" for models. Unfortunately, this section is contains a few imprecise statements and errors that hamper its usefulness as it is. First, it is stated that models "consider the Sahara Desert as one region" (272). This seems to imply, by reading the following line, that there is somehow a constant dust

source all over North Africa. This is incorrect. Basically all global climate models, including the ones cited here, simulate dust emissions as a function of wind speed and certain surface characteristics, discretized on a gridded surface with horizontal resolution around 1x1 or 2x2 degrees latitude by longitude.

We agree. Indeed, the models are more detailed than simply using the Sahara as one grid cell. This is of course was not our intention, as we wanted to emphasize the scarcity of data surrounding the Sahara. We rephrased the sentence to be more precise in its meaning and to show that eastern Africa dust fluxes were determined from distant and sparse data: *“Model evaluations of dust fluxes during the Holocene in eastern Africa are currently based on very few dispersed data points⁵², where most are located in the Atlantic Ocean and few in the Arabian Sea^{3,52,53”}.*

11. Second, when comparing the data to the model it is written that models do not account for millennial-timescale variations on dust accumulation rates (275-276). This is certainly true by definition in case of simulations in equilibrium conditions for a given climate state, e.g. the mid-Holocene at 6000 years before present. A meaningful comparison with this kind of simulations should focus on a comparable period. Thus this sentence does not provide any useful information.

The intention of this sentence is to provide examples of how exactly the models give erroneous predictions when based on sparse and distant data points. For example, the models fail to catch the dust flux variations between the early and late Holocene that are recorded in the new data we report. We rephrased it so it would be more clear:

“Consequently, these models lack the sensitivity to capture eastward Saharan dust emissions.” The meaningful comparison is given in the modified Figure 4 that depicts the two time slices in the model and in the data.

12. Third, it is said that the models simulations in question overestimate dust deposition in the Red Sea cores by an order of magnitude (e.g. Figure 4). If we compare the values after the appropriate unit conversion, we can see that this is not true.

We disagree. The conversion between [$\text{g m}^{-2} \text{yr}^{-1}$] and [$\text{g cm}^{-2} \text{ka}^{-1}$] is of one order of magnitude. Thus, model predictions are converted to 1 – 10 $\text{g cm}^{-2} \text{ka}^{-1}$ (originally reported as 10-100 $\text{g m}^{-2} \text{yr}^{-1}$; e.g., Albani et al., 2015) while our results are in the range of 0.4 – 0.9 $\text{g cm}^{-2} \text{ka}^{-1}$ (Fig 2 in the current paper). This is further depicted in figure 4 where the model predictions are higher (also note the log scale) than the data. After double-checking the unit conversion, we conclude that our original values were correct and did not change them.

13. Fourth, it is reported that models fail to capture significant differences between the late Holocene and the mid Holocene values (297-298), but it is not clear on which grounds this statement is based.

Model predictions have the same range of accumulation rates for both intervals, primarily due to the lack of data in the Eastern Africa region. We added the time slices referred to by the models in Figure 4. Thus, the reader can see the difference between the time slices.

14. Finally, the extrapolation segment for the KL15 dust flux curve from Figure 1a should be removed.

We accept this comment but note that Figure 1a is based on observations and measured data, including in KL15. The potential gap referred to by the reviewer is better identified in Figure 2a, where we indeed revised the figure and emphasized the lack of observations in KL15 between ~19-12 ka.

Reviewer #3 (Remarks to the Author):

15. Dear authors,

After reading the revised version of the manuscript, I must acknowledge that the authors are considerably improved the initial version of their manuscript, following the reviewers recommendations and comments, clarifying the points which were raising questions. On my side, I appreciate the effort to compile most available data, noticing that by now the manuscript refers to 53 papers compare to the initial 30. I have carefully checked both version and feel much more happy with the present version than with the initial one and the addition of figure 4 also clarify the authors' conclusion.

16. In the manuscript remain some minor corrections to be made like line 38, misspelling of "environments"
corrected.

17. line 223 and 234 you write "grotte de Pist" while on figure 3 it reads "Grotte de PistE". In Wassenburg et al paper it is "Grotte de Piste". So please correct in the text.
Indeed, we added the "e" to both locations.

18. line 250, when referring to earlier interglacial peaks I would rather use the chronological arrow by writing "MIS9,7, 5e. Same comment line 297, rather writing "between the mid Holocene and the late Holocene.
We thank the reviewer and made the appropriate changes in the text and figure captions.

REVIEWERS' COMMENTS:

Reviewer #2 (Remarks to the Author):

The authors have provided satisfactory responses to the comments, and I find this last version to be a fine manuscript, for which I recommend publication.

Response to Referees

REVIEWERS' COMMENTS:

Reviewer #2 (Remarks to the Author):

The authors have provided satisfactory responses to the comments, and I find this last version to be a fine manuscript, for which I recommend publication.

We thank the reviewer for his support and assistance.